# Mechanisms of Gut-Related Viral Persistence in Long COVID

**DOI:** 10.3390/v16081266

**Published:** 2024-08-07

**Authors:** Philip McMillan, Anthony J. Turner, Bruce D. Uhal

**Affiliations:** 1McMillan Research Ltd., London WC2H 9JQ, UK; philip.mcmillan@nhs.net; 2School of Biomedical Sciences, Faculty of Biological Sciences, University of Leeds, Leeds LS2 9JT, UK; a.j.turner@leeds.ac.uk; 3Department of Physiology, Michigan State University, East Lansing, MI 48824, USA

**Keywords:** long COVID, PASC, viral persistence, SARS-CoV-2, gut

## Abstract

Long COVID (post-acute sequelae of COVID-19—PASC) is a consequence of infection by SARS-CoV-2 that continues to disrupt the well-being of millions of affected individuals for many months beyond their first infection. While the exact mechanisms underlying PASC remain to be defined, hypotheses regarding the pathogenesis of long COVID are varied and include (but are not limited to) dysregulated local or systemic inflammatory responses, autoimmune mechanisms, viral-induced hormonal imbalances, skeletal muscle abnormalities, complement dysregulation, novel abzymes, and long-term persistence of virus and/or fragments of viral RNA or proteins. This review article is based on a comprehensive review of the wide range of symptoms most often observed in long COVID and an attempt to integrate that information into a plausible hypothesis for the pathogenesis of PASC. In particular, it is proposed that long-term dysregulation of the gut in response to viral persistence could lead to the myriad of symptoms observed in PASC.

## 1. Introduction

In the four years since the first appearance of COVID-19, there have been more than 774 million cases globally and a reported death rate of almost 1% from the disease [1]. These are likely underestimates given the inadequacy of detection and reporting methodologies across the world. While vaccinations and new treatment methods have helped to bring the disease under control to some degree, post-COVID-19 sequelae, a syndrome termed long COVID, has blighted the lives of many individuals. This review highlights the pathology of the chronic disease and the wide range of reported symptoms and proposes underlying mechanisms that drive disease progression and viral persistence. The key hypothesis proposed here is that long COVID (post-acute sequelae of COVID-19—PASC) is primarily a disease driven by the gut that is perturbed by underlying inflammation. 

Long COVID was first documented in the months after the COVID-19 pandemic began and has been associated with over 200 symptoms, with occurrence and severity varying among affected individuals. Long COVID can best be described as a multi-layered pathology that causes a variety of symptoms that are dependent on the baseline characteristics of the individual patient. The multi-faceted nature of this disease makes it inherently difficult to diagnose and treat. Some of the most common symptoms include fatigue, headache, myalgia, concentration deficit, and sleep abnormalities [2].

The approach taken by the authors to clarifying this complex disease is, however, not to focus on symptoms, but to identify the primary pathophysiology and explain how symptoms can be so varied. Long COVID has been described as more likely among patients infected during early pandemic waves and who developed severe disease [3]; the study of Sudre et al. [4] found that long COVID was more likely to occur in patients who reported five or more symptoms. It is important to differentiate long COVID—which may occur after either a mild or severe infection—from the impact of heart, lung, or kidney damage caused by a severe cytokine immune response to COVID-19. Patients with organ damage are more likely to have primarily specific symptoms like shortness of breath, and not the wide array of symptoms typically seen in long COVID [5,6,7,8]. People who have had severe COVID-19 are not necessarily more prone to long COVID than those who had a milder case, but the resulting damage to vital organs could lead to a longer-term impact from long COVID. The present review includes studies of patients who had relatively mild to moderate COVID-19 and who did not need hospitalization [9].

## 2. The Cellular Receptor for COVID-19 Infection: Angiotensin Converting Enzyme-2 (ACE-2)

Following the original SARS coronavirus outbreak in South Asia in 2003, the cell surface receptor for the virus was rapidly identified as ACE-2, a protein discovered only three years earlier as a homologue of ACE, a key enzyme of the renin–angiotensin system (RAS) [10,11]. Similarly, ACE-2 is the primary receptor for SARS-CoV-2 [12] and also for the coronavirus CoV-NL63, which causes the common cold. ACE-2, like ACE, is a cell-surface zinc metallopeptidase; it acts as a counter-regulatory protein to ACE in the cardiovascular system [13]. The removal of ACE-2 from the plasma membrane following COVID-19 infection therefore contributes to the cardiovascular complications seen in the disease. However, ACE-2 is a multifunctional protein also facilitating intestinal amino acid transport, particularly tryptophan, through its association with the B0AT1 amino acid transporter protein, similar to the role of the protein collectrin in the kidney, which is homologous to the C-terminal domain of ACE-2 [14].

## 3. Viral Infection Routes, ACE-2, and Autoimmunity in Severe COVID-19

Severe COVID-19 with respiratory compromise has been described as a viral-mediated autoimmune response triggered by the combination of viral spike protein and serum ACE-2. It has previously been demonstrated that serum ACE-2, shed from the cell membrane, could bind tightly to the viral spike protein and potentially become caught up in the immune presentation of the virus by antigen-presenting cells [15,16]. The immune system could then produce antibodies to ACE-2 [17] as well as to the virus. The ACE-2 autoantibodies could cause endothelial inflammation in the blood vessels of the lungs, heart, and kidneys, leading to the development of microthrombi and respiratory compromise. ACE-2 is primarily located in the lungs, intestine, heart and kidneys, testes, the endothelial lining of blood vessels, and especially the upper airways [18]. It is also located in the brain, playing a key role in central cardiovascular regulation [19]. Interestingly, a study of host genetic polymorphisms [20] showed that the ACE-2 rs2285666 T allele, among several other gene polymorphisms, was associated with an increased risk of long COVID; at present, however, the mechanisms underlying these associations are unknown.

The SARS-CoV-2 virus infects the nasopharynx through binding to ACE-2 primarily on sustentacular cells [21], which express a high level of cell-surface ACE-2. Once infected, the virus employs a number of strategies to evade the interferon response [22], allowing for further infection, and spread asymptomatically to the lungs. Viral replication allows for further infection of the lower airways, finally involving alveolar macrophages [23]. Local spread to endothelial cells of lung blood vessels [24] is the final stage prior to blood-borne dissemination of the virus, which leads to infection of intestinal cells, causing a positive stool result within the first four days of infection, and positive infection of intestinal enterocytes can also be observed [25].

## 4. Viral Persistence in COVID-19

Viral persistence is central to the pathology of long COVID, as revealed by long-term circulating levels of viral proteins in the serum of patients with ongoing symptoms [26]. It is important to note that attempted culture of these viral remnants did not reveal viable viruses; that is, there are pieces of viral protein circulating rather than a persistent viral infection. A key question is, which cells are host to these viral remnants?

There is a close association between the occurrence of long COVID and inflammatory bowel disease (IBD)—of the ~10% of IBD patients who have contracted COVID-19, up to 40% of those develop long COVID with symptoms of IBD [27]—which suggests that sites of intestinal pathology may be a primary location of viral persistence. Intestinal biopsies demonstrated concentration of viral protein, including nucleocapsid and spike proteins, in specific regions of the intestine [28]. Identifying the concentration of viral remnants in specific intestinal cells facilitates an understanding of whether a specific type of intestinal cell is the primary target.

A recent tissue study of long COVID was based on 46 patients with a background of IBD and recent SARS-CoV-2 infection. An endoscopy study was conducted with biopsies from the small and large intestines. The biopsy specimens were collected in formalin and tissue samples were further processed using SARS-CoV-2 PCR, immunofluorescence, and viral culture to analyze for antigen persistence [29].

The presence of viral RNA was detected in 31% of biopsy specimens from the duodenum, ileum, and colon, but there was no evidence of ability to culture the virus. Specific immunofluorescence demonstrated involvement of epithelial intestinal cells and CD8 T lymphocytes. It is unclear which type of epithelial cell is primarily involved. The intestinal lining is primarily composed of absorptive enterocytes covering the lamina propria and muscularis layers. An understanding of primary cell involvement can provide clues about the associated pathophysiology in long COVID patients. Potential cell candidates are considered in Figure 1 below.

Figure 1 illustrates the various cell types, of which enterocytes are the most common in the intestinal lining and have the primary function of nutrient absorption. The biopsy review [28], however, revealed that the concentration of viral particles was localized to specific cells in the small intestine and colon as opposed to the diffuse pattern that would be expected if enterocytes were involved. Additionally, enterocytes are typically present for only about five days before being replaced by new ones, which makes them a less likely site for long-term concentration of viral remnants [29]. A more probable candidate is an intestinal cell with the ability to survive for longer periods of time.

Stem cells, which are the progenitor cells for most of the intestinal cells, are typically located in the intestinal crypts and help to maintain normal gut function. The location of these cells appears not to fit the observed pattern of localization from the intestinal samples. Additionally, if the stem cells played a key role, the viral remnants would likely be in multiple types of progenitor cells, which has not been demonstrated.

The microfold or M cells, which sample intestinal fluid to identify pathogens, are a possible source of viral persistence. However, they are typically localized to Peyer’s patches, which are immune-sensing regions in the intestine, rather than being diffusely spread through the intestinal lining [31,32,33,34]. Paneth cells modulate the microbiome and mediate the inflammatory response [35,36]. They are longer lived than enterocytes, with a lifespan of up to 60 days [37], and represent a potential source of SARS-CoV-2 viral remnants.

Similar to M cells are cup cells, which are another unique group of intestinal cells, but they are not localized on Peyer’s patches. Critically, these cells express vimentin G protein receptors, which are normally involved in sensing, and vimentin is reported to be a co-receptor facilitating SARS-CoV-2 cell entry. Extracellular vimentin could, therefore, be a target against viral invasion [38,39]. The occurrence of persistent G protein receptor antibodies in long COVID suggests that cup cells could be relevant to viral persistence.

Enterochromaffin cells (ECs) are neuroendocrine cells within the intestinal lining. They are a diverse collection of cells comprising less than 1% of intestinal cells and are primarily secretors of gut peptides that influence the intestine and distant organs [40]. They are involved in the control of gut inflammation and motility.

Finally, tuft cells are secretory epithelial cells in the intestine with the ability to sense the surrounding environment and produce biogenic amines and peptides such as serotonin and calcitonin gene-related peptide (CGRP).

The involvement of T cells in all biopsy specimens has been noted, specifically CD8 T cells in regions of active intestinal inflammation. The persistence of viral antigen has been demonstrated in CD8 pharyngeal cells [41], which could account for the persistence seen within intestinal CD8 cells. If T lymphocytes are found to be central to long COVID, then the primary pathophysiology should point to a mechanism that is consistent across all patterns of the disease.

Autopsy studies have also shed some light on the issue of viral persistence in long COVID. Stein et al. [42] looked at 44 autopsy specimens with a focus on the burden of infection, both acutely and over seven months following symptom onset. The data showed a wide distribution of viral RNA in multiple pulmonary and extrapulmonary tissues for up to 230 days following symptom onset, but with little inflammation or evidence of direct tissue or organ system injury, nor any apparent direct viral cytopathic effect outside of the lungs. These results were taken to indicate that the virus can survive and thrive in the body without generating a direct inflammatory response and suggested that the immune response is less capable outside of the lungs. The authors detected sgRNA (subgenomic RNA) in at least one tissue in over half of cases (14/27) beyond day 14, suggesting that prolonged viral replication may occur in extra pulmonary tissues as late as day 99. Outside the lungs, histological changes were mainly related to complications of therapy or pre-existing co-morbidities, mainly obesity, diabetes, and hypertension. In the 44 cases, RNA was detected in 79 of 85 anatomic locations with at least 100 N (nucleocapsid) gene copies/ng of RNA from every tissue. The authors also recovered replication-competent virus from the thalamus. In the hypothalamus, neurons were stained with anti-S and anti-N antibodies. Thus, non-intestinal and non-epithelial cells also serve as reservoirs of replicating viruses.

## 5. The Gut–Brain Axis: Potential Role in COVID-19-Related Cognitive Dysfunction

One of the symptoms of long COVID has been described as “brain fog”—difficulty in concentrating or cognitive decline. In light of the growing literature describing the now well-established concept of the “Gut-Brain Axis” [43], it can be hypothesized that gut dysfunction due to SARS-CoV-2 could be responsible for at least some of the cognitive dysfunction seen in long COVID. Indeed, Chen et al. [44] documented an association between gut microbiota dysbiosis and poor functional outcomes in acute ischemic stroke patients with COVID-19 infection. The potential crosstalk between COVID-19 and the gut–brain axis has been discussed critically [45], and Plummer et al. [46] have proposed that a hypothetical scenario in gut–brain axis pathogenesis might invoke mechanisms leading to neurocognitive decline, such as decreased intestinal and brain barrier functions that promote immune-mediated systemic and neural inflammation. On the other hand, Wong et al. [47] showed that serotonin is reduced in long COVID, and reasoned that serotonin reduction might impede the activity of the vagus and thereby impair hippocampal function and memory. Regardless, in light of the growing literature, it seems reasonable to speculate that gut dysfunction caused by SARS-CoV-2 persistence in the gut could contribute to the neurocognitive decline in PASC.

## 6. Gut Autoimmunity

How is the intestine predisposed to persistent viral replication, and is it the main viral reservoir in long COVID? There are higher concentrations of EC in coeliac disease and IBS [48]. If these cells are the primary route to the intestinal infection, higher numbers of ECs with basolateral B0AT1 transporters could increase the risk of this infection occurring. Basolateral B0AT1 transporters with ACE-2 have not been identified, but there are high numbers on the apical side to allow for probable infection [49]. Additionally, fecal microbiome changes point strongly to SARS-CoV-2 infection as inducing an immune response in the gut [50].

Immune responses related to food intolerance in the gut are tightly regulated, primarily through T cell activation, as demonstrated in coeliac disease [51]. Intestinal infection with SARS-CoV-2 can dysregulate the balance between regulatory and effector T cells and potentially increase intestinal inflammation [52]. The association between SARS-CoV-2 infection and lymphopenia suggests an immune response in the gut [53] where cell-mediated immunity is a pre-existing issue.

## 7. Viral Persistence in Specific Cell Types

This paper is focused on intestinal ECs, tuft cells, and cup cells as potential primary locations for viral remnants related to SARS-CoV-2 and as possible drivers for persistent immune responses in long COVID.

EC cells have serotonin as the primary biogenic amine and account for over 95% of total body serotonin content. EC uptake of its precursor, tryptophan, from the intestinal lumen is likely to be facilitated by the B0AT1 amino acid transporter with associated ACE-2 acting as a chaperone protein, like its homologue, collectrin [14].

SARS-CoV-2 serum viremia could target the intestinal ECs for infection. There is evidence that SARS-CoV-2 can broadly infect intestinal cells with ongoing diffuse replication, suggesting luminal enterocyte infection [54]. In the context of serum viremia, it is probable that ECs are one of the primary routes to gaining access to the intestinal lumen. Once in intestinal secretions and fecal matter, viral particles can continue to replicate in the latter [55] and subsequently infect enterocytes, which have ACE-2 concentrated on the luminal portion of the cell. Prolonged fecal shedding of viral RNA for up to 210 days post-infection points to the intestine as the source of ongoing viremia [56].

Tuft cells are potentially associated with long COVID symptoms. It has already been established that they have the characteristic of viral persistence as norovirus [57]. The tuft cells, once infected, are able to evade clearance from the immune system.

Additionally, tuft cells have the unique ability to be chemosensing with regard to sweet and bitter [58]. Chemosensing takes place through a G protein receptor. It is possible that interaction with bacterial toxins through persistent infection [59] could stimulate an immune response to these G protein receptors. Persistently elevated autoantibodies to G protein receptors could also explain many of the diverse changes in long COVID [59,60,61]. The association of G protein receptors with chemosensing intestinal cells could also be linked to taste and smell abnormalities, which are some of the most common symptoms in long COVID [62].

Viral antigens were definitively located in CD8+T cells within the intestinal lining. What was not clear was the exact morphology of these CD8+ cells. One group of CD8+ cells are the mucosal-associated invariant T (MAIT) cells, which have unique characteristics such as the ability to bridge innate and adaptive immunity [63]. Interaction with intestinal microbes and concentration in the intestinal tract could constitute a trigger mechanism for inflammation [64]. These cells circulate throughout the body and thus could be relevant to the broad spectrum of symptoms associated with long COVID [65].

## 8. Mosaic Theory of Long COVID

The spectrum of disease manifestations in long COVID is broad. The immune response to persistence of SARS-CoV-2 viral remnants in specific intestinal cells is likewise expected to vary and will depend on the number and type of cells involved as well as the predisposition of the individual. However, patterns are likely to evolve, which are relatively consistent. This understanding has led to the formulation of a mosaic theory of long COVID. The mosaic theory has been applied elsewhere in medical research; for example, in hypertension, it was proposed that interaction of many factors led to elevated blood pressure and end-organ damage [66].

Smell and taste abnormalities in the disease suggest involvement of tuft cells as a main driver for some COVID-19 symptoms; a study by Santin et al. [67] suggested the possible involvement of genetic variants of the extraoral bitter (TAS2Rs) and sweet (TAS1Rs) taste receptors. Cup cells with associated vimentin could be related to G protein receptor autoantibodies, potentially leading to low cortisol in long COVID [68] and associated fatigue through potential impact on the G protein cortisol feedback receptors in the pituitary. Infected EC involvement might cause postural orthostatic tachycardia syndrome (POTS)-type symptoms and depression through serotonin interaction, since POTS is a multisystem disorder frequently initiated by viral infection [69]. Up to 60% of COVID-19 patients can develop POTS-like symptoms [70].

Symptoms of long COVID fatigue are very similar to post-viral fatigue syndrome [71]. On the other hand, some subjective yet specific symptoms such as chronic fatigue, myalgia, concentration deficit, and insomnia should be screened for false positives, especially in patients developing mild to moderate disease [3]. It has been noted that reactivation of Epstein–Barr virus (EBV) is associated with COVID-19 infection [72] and could be a contributor to some of the patterns in the disease. The association with EBV gut persistence and gut immunity [73] is especially important in the context of SARS-CoV-2 viral persistence in the gut immune system.

Coagulation activation and microclots are a typical feature of long COVID and highlight an additional facet of the disease. Indeed, Cervia-Hassler et al. [74] analyzed complement system components and biomarkers in long COVID and identified a series of anomalies described as complement system dysregulation and ongoing activation of both alternative and classical complement pathways.

Elevated levels of von Willebrand factor (VWF) in long COVID along with low levels of the metalloprotease ADAMTS13 could point to an intestinal toxin as an associated factor [75]. In shigellosis, the exotoxin from the bacterial infection can lead to excessive clotting in the form of thrombotic thrombocytopenic purpura [76]. The parallel with toxin-related microclots in long COVID is striking. In consideration of all these findings, it is proposed (see Figure 2) that at least three key factors can interact to promote the occurrence of severe long COVID: EBV reactivation, gut immunity, and the occurrence of microclots.

## 9. Discussion

Long COVID appears to be a viral-mediated inflammatory disease with dysregulated intestinal immune responses as the primary driver of symptoms. The predisposition for long COVID appears to be prior chronic intestinal inflammation. The inflamed areas have the potential for increases in specific cell types, such as ECs, tuft cells, and cup cells, and could be the location of viral remnants that cause ongoing symptoms.

Long COVID has a heterogeneous pattern that depends on the characteristics of prior gut inflammation and whether it is T cell or humoral mediated. Overall, the SARS-CoV2 viral infection may cause dysregulated T cell control in inflamed areas of the intestine, exacerbating the chronic inflammation. The heterogeneity of long COVID indicates that multiple factors could be at play. With a more detailed understanding of the symptoms, specific cell types may be targeted to reduce the probability of viral persistence. Broad symptoms suggest there are a number of overlapping factors at play that make it difficult to identify primary pathology. Bacterial toxins could be a contributing factor in long COVID; they are known to be a part of other diseases such as chronic fatigue syndrome and fibromyalgia and could be related to immune responses to bacterial exotoxins.

To be fair, other authors including Appelman et al. [77] have interesting data to suggest that infiltration of amyloid deposits and inflammatory cells into skeletal muscle may have a role to play in long COVID pathogenesis, at least with respect to the post-exertional malaise often observed in these patients. Another interesting study by Song et al. [78] reported the finding of ACE-2-like enzymatic activity within immunoglobulins (“abzymes”) in the serum of patients with long COVID. Given that ACE-2 degrades angiotensin II and produces the peptide angiotensin 1–7, which itself has interesting physiologic roles [15,16], the potential role(s) of these abzymes in long COVID pathogenesis cannot be discounted.

An important but poorly understood issue is the ability of COVID-19 vaccinations to prevent the long-term symptoms often observed in PASC. In a staggered cohort study of COVID-19 patients in the UK, Spain, and Estonia, Catala et al. [79] found that both mRNA-based and adenoviral-based COVID-19 vaccines were effective at reducing the risk of long COVID symptoms as defined by the WHO. The mechanism(s) by which vaccination reduces the risk of long COVID symptoms are at present unknown, but deserve further attention, particularly in light of prior vaccine hesitancy due to safety concerns and the lower risk of severe COVID-19 in young people (ibid).

On the other hand, a decreased risk of long COVID symptoms was reported not only following vaccination, but also after SARS-CoV-2 reinfections, suggesting the role of pre-existing immunity acquired by either natural infection, vaccination, or both (hybrid) to prevent post-COVID-19 sequelae [3]. In fact, LC dropped dramatically after reinfections, which almost entirely surged during omicron [3,80], although rare reinfections observed during the first 1–2 pandemic years were reportedly more severe than the respective primary COVID-19 events.

Clarification of the characteristics of the immune response could direct the appropriate immune-suppressive approach to manage long COVID. The evidence strongly points to long COVID being a complex, multisystemic viral-mediated autoimmune disease. Developing comprehensive answers to the questions raised above will undoubtedly take considerable time and further research to resolve the origins and potential therapies for long COVID.

## Figures and Tables

**Figure 1 viruses-16-01266-f001:**
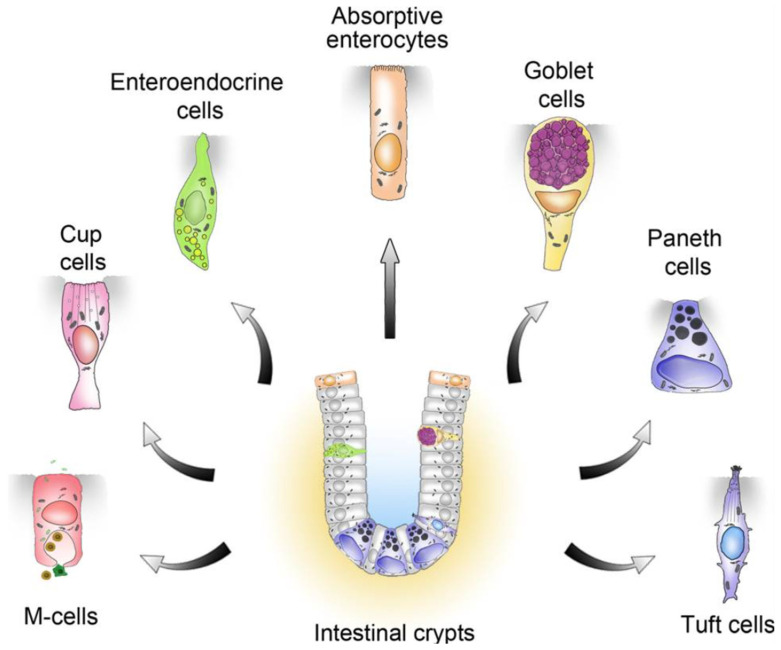
Intestinal epithelial cell types (reprinted with permission from Gerbe et al., 2012 [30]).

**Figure 2 viruses-16-01266-f002:**
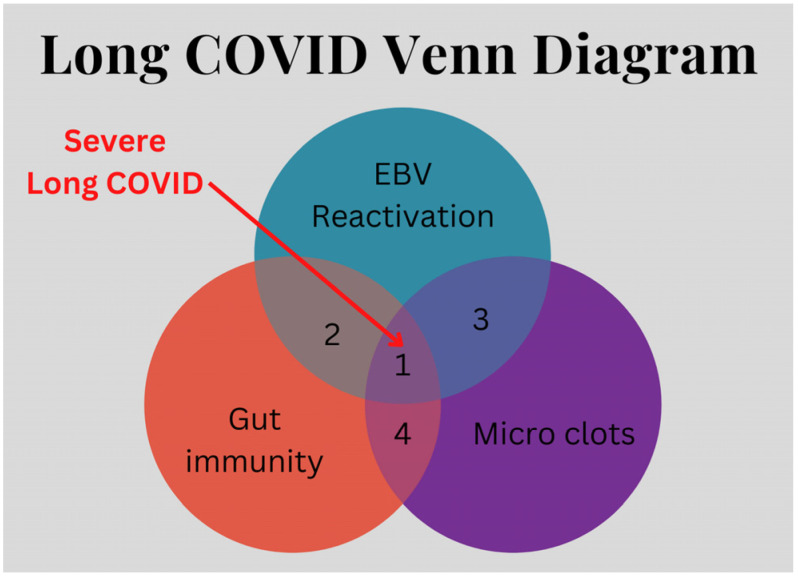
Long COVID is a combination of multiple diseases.

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
