# Peer review of "Mechanisms of Gut-Related Viral Persistence in Long COVID"

_viruses, 2024, doi:10.3390/v16081266_

Round 1
Reviewer 1 Report
Comments and Suggestions for Authors
General feed back
This narrative review examined the available evidence to advanced an hypothesis on dysregulated intestinal immune responses as the primary driver of long-COVID-19, especially in presence of pre-existing chronic intestinal inflammation.
In particular, the authors narrowed down their speculative hypothesis on entero-chromaffin, tuft and cup cells as potential primary locations for viral remnants related to SARS-CoV2 persistence outside the lung.
Specific comments
· The paper presents LC as a current issue, failing to stress that this is a syndrome almost exclusively regarding the past, i.e. initial phase of the pandemic, when more virulent viral strains were circulating, vaccination was not available yet and severe cases of COVID-19 were more frequent. Now COVID-19 and post-COVID-19 sequelae can be a problem almost entirely for patients at high risk of developing severe disease. It is in fact expected that following an infectious disease there is a period of convalescence before recovery, and this period increases with the severity of the acute disease and hospital admission [recommended citations: PMID: 38140174; PMID: 36474804]. However, the pandemic scenario changed dramatically at the end of 2021, when Omicron spread and population immunity against COVID-19 developed - by either natural infection and/or vaccination - . With Omicron the hospitalization rate dropped to <0.3% ( https://www.ecdc.europa.eu/en/covid-19/latest-evidence/clinical). Moreover, the vast majority of patients with post-COVID-19 tended to return to a state of full health with time [PMID: 36676046].
· Although there is evidence that the risk of long-COVID-19 increases with viral shedding time from the nasal cavity [PMID: 38140174], persistence of SARS-CoV-2 needs to be taken with caution. Localized persistence of infection in the nasal cavity dose in patients developing mild/asymptomatic disease does not necessarily mean neurovascular invasion with systemic effects. Moreover, patients with pre-existing psychological conditions (e.g. depression, anxiety) were more likely to report long-COVID-19 [PMID: 36676046], thereby questioning the reliability of some of their subjective post COVID19 symptoms (fatigue, concentration deficit, myalgia, insomnia, among other) [recommended citation: PMID: 38140174, PMID: 34631916]. Since post COVID-19 symptoms (especially asthenia, but also other subjective conditions as myalgia or concentration deficit) could be easily prone to false positive results, they should be adequately screened, especially in those developing mild-asymptomatic disease [PMID: 34631916; PMID: 38140174].
· Line 26: “long-term persistence of the disease,” change to post-COVID-19 sequelae, as these include both symptoms persisting since acute disease and newly developed after the end of acute disease (or following a negative swab test result)
· Line 38-39: myalgia and concentration deficit to be included
· Line 42-43: again, this is a questionable point, since there is evidence that long-COVID-19 is far more likely among patients infected during the early pandemic waves and developing severe disease [recommended citation: PMID: 38140174].
· Line 198-100: “The association with SARS-CoV2 infection and lymphopenia”… the association of what ?
· Line 783-84: upper airways secondary location of ACE-2? That seems contradicting line 77, reporting that nasopharynx “expresses a high level of cell-surface ACE-2”.
· Line 90-100: how recent was the infection?
· Line 138: “It could”… it= vimenin?
· Lie 248. It is unclear how EC cells involvement would play a role in the persistence of post-COVID-19 dyspnea?
· Line 252-255: some subjective yet aspecific symptoms as chronic fatigue, myalgia, concentration deficit, insomnia should be screened for false positives, especially in patients developing mild-to moderate disease [recommended citation: PMID: 38140174].
· Line 284-291: a decreased risk of long-covid-19 symptoms was reported not only following vaccination, but also after SARS-Cov-2 reinfections, suggesting the role of pre-existing immunity acquired by either natural infection, vaccination or both (hybrid) to prevent post-COVID-19 sequelae [recommended citation: PMID: 38140174]. LC dropped in fact dramatically after reinfections, which almost entirely surged during Omicron [recommended citations: PMID: 38140174; PMID: 37515237], although rare reinfections observed during the first 1-2 pandemic years were reportedly more severe than the respective primary COVID-19 events.
Author Response
Response: We thank the reviewer for the astute and constructive criticisms. In response, we added 9 new citations suggested by both reviewers, and associated discussions of each new reference as they relate to the original text. We attempted to address all concerns of the reviewer at the text lines suggested for each point.
All revisions are in red font. Responses to the main issues raised by the reviewer, i.e. risk factors for developing long COVID and symptomology, we now discuss these issues in lines 42-46, 77-82 and 308-314. We hope that reviewer finds these revisions acceptable.
Reviewer 2 Report
Comments and Suggestions for Authors
The manuscript addresses various aspects of COVID-19, including mechanisms, risk factors, management of long COVID, and the interplay between COVID-19 and the gut-brain axis. It references several studies and reviews that discuss the long-term effects of COVID-19, the role of angiotensin-converting enzyme (ACE2), and the implications of chronic stress in the context of the gut-brain-immune axis. Overall, the manuscript is suitable for publication. It provides a thorough and well-supported overview of long COVID, making it a valuable contribution to the field. The authors have cited relevant and up-to-date studies, ensuring that their discussion is grounded in the latest research.
Author Response
Response: We thank the reviewer for the astute and constructive criticisms. In response, we added 9 new citations suggested by both reviewers, and associated discussions of each new reference as they relate to the original text. We attempted to address all concerns of the reviewer at the text lines suggested for each point.
All revisions are in red font. Responses to the main issues raised by the reviewer, i.e. i.e. risk factors for developing long COVID and symptomology, we now discuss these issues in lines 42-46, 77-82 and briefly elsewhere. We hope that reviewer finds these revisions acceptable.
Reviewer 3 Report
Comments and Suggestions for Authors
Macmillan et al wrote a review about the possible reason of developing symptoms of Long Covid-19 syndrome by the gut microbial association of the patient COVID-19 patients. Although the exact mechanisms of Long COVID are not known, several reasons are speculated from RNA seq, single cell RNA seq and molecular analysis from blood, brain and lung tissues (autopsy) of Long COVID-19 patients. There are also many reports of specific association of gut microbiota with Long COVID. Here authors discuss the specific intestinal cells that could cause Long Covid symptoms. Although the specific characterization of cells is interesting, authors ignored the rigorous critical evaluation of developing Long COVID phenotypes.
These are the following comments.
1. Line 46-48, many scientists believe that severity of COVID-19 is the risk factor for Long COVID. Here, authors stated that mild or moderate COVID-19 has equal probability of Long CoVID. However, are there any symptomatic differences of Long COVID developed from mild or deVere COVID-19 (Authors should discuss this with reference (if available)
2. In introduction, authors should discuss another reasons/hypothesis for long COVID syndrome. They should consider these papers. doi: 10.1038/s41467-023-44432-3 DOI: 10.1126/science.adg7942; DOI: 10.1128/mbio.00541-24
3. Line 51-84, They should strengthen these with this report where ACE2 host mutation is identified in long COVID patients. DOI: 10.1080/22221751.2023.2239952
4. In line 244, for taste abnormalities, this paper could be discussed doi:10.3390/life14020219
5. Line 146-147, Authors should incorporate and discuss this paper in the context of serotonin production and release in defect in Long COVID patients DOI: 10.1016/j.cell.2023.09.013
6. Author should accommodate other reasons/hypothesis in the mosaic theory along with the role of gut microbiota as the disease is very heterogenous.

Author Response
We thank the reviewer for all the constructive comments and suggestions. In total, we now added 9 new citations to address all the reviewers concerns, with revisions at the line numbers indicated by the reviewer. Concerns about risk factors for long COVID are addresses by new text at lines 42-47 and 77-82. All revisions, numerous thorught the paper, are in red font. We hope these revisions are now acceptable.
Round 2
Reviewer 1 Report
Comments and Suggestions for Authors
The authorss addressed comments from peer review.
Author Response
We thank the reviewer for the positive assessment
Reviewer 3 Report
Comments and Suggestions for Authors
The review is much improved. However, authors did not discuss the other possible explanations of developing Long COVID-19 symptoms, especially neuronal, cardiovascular abnormalities and abzyme hypothesis in the introduction or in discussion (mentioned/reference in earlier (1st review) comments).
Author Response
Comment 1: The review is much improved. However, authors did not discuss the other possible explanations of developing Long COVID-19 symptoms, especially neuronal, cardiovascular abnormalities and abzyme hypothesis in the introduction or in discussion.
Response: We thank the reviewer for alerting us to the pertinent references describing alternate theories for long COVID pathogenesis. We now highlite these in the abstract (lines 15-16) and discuss them, with citations, in lines (275-278, 305-308 and 308-313).
We hope the improved manuscript is now acceptable.